# AN EXPLICITLY RELATIONAL NEURAL NETWORK ARCHITECTURE

## ABSTRACT

With a view to bridging the gap between deep learning and symbolic AI, we present a novel end-to-end neural network architecture that learns to form propositional representations with an explicitly relational structure from raw pixel data. In order to evaluate and analyse the architecture, we introduce a family of simple visual relational reasoning tasks of varying complexity. We show that the proposed architecture, when pre-trained on a curriculum of such tasks, learns to generate reusable representations that better facilitate subsequent learning on previously unseen tasks when compared to a number of baseline architectures. The workings of a successfully trained model are visualised to shed some light on how the architecture functions.

## 1 INTRODUCTION

When humans face novel problems, they are able to draw effectively on past experience with other problems that are superficially very different, but that have similarities on a more abstract, structural level. This ability is essential for lifelong, continual learning, and confers on humans a degree of data efficiency, powers of transfer learning, and a capacity for out-of-distribution generalisation that contemporary machine learning has yet to match (Garnelo et al., 2016; Lake et al., 2017; Marcus, 2018; Smith, 2019). A case may be made that all these issues are different facets of the same underlying challenge, namely the challenge of devising systems that learn to construct *general-purpose, reusable representations* (McCarthy, 1987; Bengio et al., 2013). A representation is general-purpose and reusable to the extent that it contains information whose domain of application exceeds the context within which it was acquired.

Representations that are general-purpose and reusable improve data efficiency because a system that already knows how to build representations relevant to a novel task (despite its novelty) doesn't have to learn that task from scratch. Ideally, a system that efficiently exploits general-purpose, reusable representations in this way should be the very same system that learned how to construct them in the first place. Moreover, in learning to solve a novel task using such representations, we should expect the system to learn further representations that are themselves general-purpose and reusable. So, with the exception of the very first representations the system learns, all learning in such a system would in effect be transfer learning, and the process of learning would be inherently cumulative, continual, and lifelong.

One approach to building such a system is to take inspiration from the paradigm of classical, symbolic AI (Garnelo & Shanahan, 2019). Building on the mathematical foundations of first-order predicate calculus, a typical symbolic AI system works by applying logic-like rules of inference to language-like propositional representations whose elements are objects and relations. Thanks to their declarative character and compositional structure, these representations lend themselves naturally to generality and reusability. However, in contrast to contemporary deep learning systems, the representations deployed in classical AI are not usually learned from data but hand-crafted (Harnad, 1990). The aim of the present work is to get the best of both worlds with an end-to-end differentiable neural network architecture that builds in propositional, relational priors in much the same way that a convolutional network builds in spatial and locality priors.

The architecture introduced here builds on recent work with non-local network architectures that learn to discover and exploit relational information (Wang et al., 2018), notably relation nets (Santoro et al., 2017; Palm et al., 2018) and architectures based on multi-head attention (Vaswani et al., 2017;

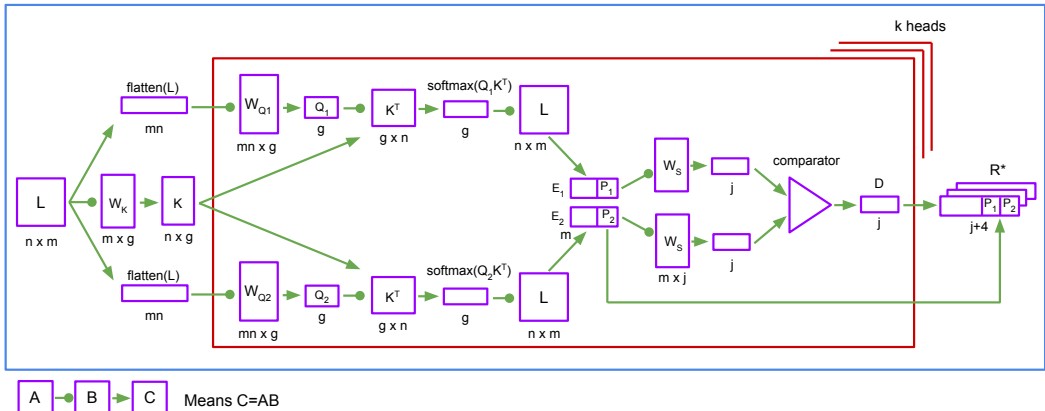

Figure 1: The PrediNet architecture. $W_K$ and $W_S$ are shared across heads, whereas $W_{Q1}$ and $W_{Q2}$ are local to each head. See main text for more details.

Santoro et al., 2018; Zambaldi et al., 2019). However, these architectures generate representations that lack explicit structure. There is, in general, no straightforward mapping from the parts of a representation to the usual elements of a symbolic medium such as predicate calculus: propositions, relations, and objects. To the extent that these elements are present, they are smeared across the embedding vector, which makes representations hard to interpret and makes it more difficult for downstream processing to take advantage of compositionality.

Here we present an architecture, which we call a PrediNet, that learns representations whose parts map directly onto propositions, relations, and objects. To build a sound, scientific understanding of the proposed architecture, and to facilitate a detailed comparison with other architectures, the present study focuses on simple tasks requiring relatively little data and computation. We develop a family of small, simple visual datasets that can be combined into a variety of multi-task curricula and used to assess the extent to which an architecture learns representations that are general-purpose and reusable. We report the results of a number of experiments using these datasets that demonstrate the potential of an explicitly relational network architecture to improve data efficiency and generalisation, to facilitate transfer, and to learn reusable representations.

The main contribution of the present paper is a novel architecture that learns to discover objects and relations in high-dimensional data, and to represent them in a form that is beneficial for downstream processing. The PrediNet architecture does not itself carry out logical inference, but rather extracts relational structure from raw data that has the potential to be exploited by subsequent processing. Here, for the purpose of evaluation, we graft a simple multi-layer perceptron output module to the PrediNet and train it on a simple set of spatial reasoning problems. The aim is to acquire a sufficient scientific understanding of the architecture and its properties in this minimalist setting before applying it to more complex problems using more sophisticated forms of downstream inference.

## 2 THE PREDINET ARCHITECTURE

The idea that propositions are the building blocks of knowledge dates back to the ancient Greeks, and provides the foundation for symbolic AI, via the $19^{\text{th}}$ century mathematical work of Boole and Frege (Russell & Norvig, 2009). An elementary proposition asserts that a relationship holds between a set of objects. Propositions can be combined using logical connectives (and, or, not, etc), and can participate in inference processes such as deduction. The task of the PrediNet is to (learn to) transform high-dimensional data such as images into propositional representations that are useful for downstream processing. A PrediNet module (Fig. 1) can be thought of as a pipeline comprising three stages: *attention*, *binding*, and *evaluation*. The attention stage selects pairs of objects of interest, the binding stage instantiates the first two arguments of a set of three-place predicates (relations) with selected object pairs, and the evaluation stage computes values for each predicate's remaining (scalar) argument such that the resulting proposition is true.

More precisely, a PrediNet module comprises $k$ heads, each of which computes $j$ relations between pairs of objects (Fig. 1). The input to the PrediNet is a matrix, $L$, comprising $n$ rows of feature vectors, where each feature vector has length $m$. In the present work, $L$ is computed by a convolutional neural network (CNN). The CNN outputs a feature map consisting of $n$ feature vectors that tile the input image. The last two elements of the feature vector are the xy co-ordinates of the associated patch in the image. So the length $m$ of each feature vector corresponds to the number of filters in the final CNN layer plus 2 (for the co-ordinates), and the $i^{th}$ element of a feature vector (for $i < m - 2$) is the output of the $i^{th}$ filter. For a given input $L$, each head $h$ computes the same set of relations (using shared weights $W_S$) but selects a different pair of objects, using dot-product attention based on key-query matching (Vaswani et al., 2017). Each head computes a separate pair of queries $Q_1^h$ and $Q_2^h$ (via $W_{Q1}^h$ and $W_{Q2}^h$). But the key space $K$ (defined by $W_K$) is shared, so that the set of entities that are candidates for attention is consistent across heads. The whole (flattened) image is used to generate queries, allowing attention masks to depend on the image's full (non-local) content.

$$Q_1^h = \text{flatten}(L)W_{Q1}^h \qquad Q_2^h = \text{flatten}(L)W_{Q2}^h \qquad K = LW_K$$

Applying the resulting pair of attention masks directly to $L$ yields a pair of objects $E_1^h$ and $E_2^h$, each represented by a weighted sum of feature vectors.

$$E_1^h = \text{softmax}(Q_1^h K^\top)L \qquad\qquad E_2^h = \text{softmax}(Q_2^h K^\top)L$$

All $j$ relations between $E_1^h$ and $E_2^h$ are then evaluated. There are many ways to compute a relationship between a pair of objects represented as feature vectors. We chose to compute the values of relations by taking vector differences, which has been shown to be effective in the context of relationally structured knowledge bases (Bordes et al., 2011; Socher et al., 2013). In the current architecture, $E_1^h$ and $E_2^h$ are subject to a linear mapping (via $W_S$) into $j$ 1D spaces, one per relation, and the resulting vector is passed through an element-wise comparator, yielding a vector of differences $D^h$.

$$D^h = E_1^h W_S - E_2^h W_S$$

The last two elements of $E_1^h$ and $E_2^h$ (the positions $P_1^h$ and $P_2^h$, respectively) are concatenated to the vector $D^h$ of differences to give the head's output $R^h = (D^h, P_1^h, P_2^h)$. Finally, the outputs of all $k$ heads are concatenated, yielding the output of the PrediNet module, a vector $R^*$ of length $k(j + 4)$. In predicate calculus terms, the final output of a PrediNet module with $k$ heads and $j$ relations represents the conjunction of elementary propositions

$$\Psi \equiv \bigwedge_{h=1}^{k} \bigwedge_{i=1}^{j} \psi_i(d_i^h, e_1^h, e_2^h) \tag{1}$$

where $\psi_i(d_i^h, e_1^h, e_2^h)$ asserts that $d_i^h$ is the distance between objects $e_1^h$ and $e_2^h$ in the 1D space defined by column $i$ of the weight matrix $W_S$, and the denotations of $e_1^h$ and $e_2^h$ are captured by the vectors $Q_1^h$ and $Q_2^h$ respectively, given the key-space defined by $K$.

Equation 1 supplies a semantics for the PrediNet's final output vector $R^*$ that maps each of its elements onto a well-defined logical formula, something that cannot be claimed for other architectures, such as the relation net or multi-head attention net. In the experiments reported here, only $R^*$ is used for downstream processing, and this vector by itself doesn't have the logical structure described by Equation 1. However, the PrediNet module can easily be extended to deliver an additional output in explicitly propositional form, with a predicate-argument structure corresponding to the RHS of Equation 1. In the present paper, the pared-down vector form facilitates our experimental investigation, but in its explicitly propositional form, the PrediNet's output could be piped directly to (say) a Prolog interpreter (Fig.7), to an inductive logic programming system, to a statistical relational learning system, or indeed to another differentiable neural module.

## 3 DATASETS AND TASKS

It would be premature to apply the PrediNet architecture to rich, complex data before we have a basic understanding of its properties and its behaviour. To facilitate in-depth scientific study, we need small, simple datasets that allow the operation of the architecture to be examined in detail and the fundamental premises of its design to be assessed. Our experimental goals in the present paper are 1)

to test the hypothesis that the PrediNet architecture learns representations that are general-purpose and reusable, and 2) insofar as this is true, to investigate why. Existing datasets for relational reasoning tasks, such as CLEVR (Johnson et al. (2017)) and sort-of-CLEVR (Santoro et al. (2017)), were ruled out because they include confounding complexities, such as occlusion and shadows or language input, and/or because they don't lend themselves to the fine-grained task-level splits we required. Consequently, we devised a new configurable family of simple classification tasks that we collectively call the Relations Game.

A Relations Game task involves the presentation of an image containing a number of objects laid out on a $3 \times 3$ grid, and the aim (in most tasks) is to label the image as True or False according to whether a given relationship holds among the objects in the image. While the elementary propositions learned by the PrediNet only assert simple relationships between pairs of entities, Relations Game tasks generally involve learning compound relations involving multiple relationships among many objects. The objects in question are drawn from either a training set or one of two held-out sets (Fig. 2a). None of the shapes or colours in the training set occurs in either of the held-out sets. The training object set contains 8 uniformly coloured pentominoes and their rotations and reflections (37 shapes in all) with 25 possible colours. The first held-out object set contains 8 uniformly coloured hexominoes and their rotations and reflections (46 shapes in all) with 25 possible colours, and the second held-out object set contains only squares, but with a striped pattern of held-out colours.

Each Relations Game task is tied to a given relation. Even with such a simple setup, the number of definable relations among all possible combinations of objects is astronomical ($2^{(n+1)^9}$ for $n$ distinct objects), although only a few of them will make intuitive sense. For the present study, we defined a handful of intuitively meaningful relations and generated corresponding labelled datasets comprising 50% positive and 50% negative examples. A selection is shown in Fig. 2c. The 'between' relation holds iff the image contains three objects in a line in which the outer two objects have the same shape, orientation, and colour. The 'occurs' relation holds iff there is an object in the bottom row of three objects that has the same shape, orientation, and colour as the (single) object in the top row. The 'same' relation holds iff the image contains two objects of the same shape, orientation, and colour. In each case, we balanced the set of negative examples to ensure that "tricky" images involving pairs of objects with the same colour but different shape or the same shape but different colour occur just as frequently as those with objects that differ in both colour and shape.

## 4  EXPERIMENTAL SETUP

At the top level, each architecture we consider in this paper comprises 1) a single convolutional input layer (CNN), 2) a central module (which might be a PrediNet or a baseline), and 3) a small output multi-layer perceptron (MLP) (Fig. 3). A pair of xy co-ordinates is appended to each CNN feature vector, denoting its position in convolved image space and, where applicable, a one-hot task identifier is appended to the output of the central module. For most tasks, the final output of the MLP is a one-hot label denoting True or False. The PrediNet was evaluated by comparing it to several baselines: two MLP baselines (MLP1 and MLP2), a relation net baseline (Santoro et al., 2017) (RN), and a multi-head attention baseline (Vaswani et al., 2017; Zambaldi et al., 2019) (MHA).

To facilitate a fair comparison, the top-level schematic is identical for the PrediNet and for all baselines (Fig. 3). All use the same input CNN architecture and the same output MLP architecture, and differ only in the central module. In MLP1, the central module is a single fully-connected layer with ReLu activations, while in MLP2 it has two layers. In RN, the central module computes the set of all possible pairs of feature vectors, each of which is passed through a 2-layer MLP; the resulting vectors are then aggregated by taking their element-wise means to yield the output vector. Finally, MHA comprises multiple heads, each of which generates mappings from the input feature vectors to sets of keys $K$, queries $Q$, and values $V$, and then computes $\mathrm{softmax}(QK^\top)V$. Each head's output is a weighted sum of the resulting vectors, and the output of the MHA central module is the concatenation of all its heads' outputs. The PrediNet used here comprises $k = 32$ heads and $j = 16$ relations (Fig. 1). All reported experiments were carried out using stochastic gradient descent, and all results shown are averages over 10 runs. Further experimental details are given in the Supplementary Material, which also shows results for experiments with different numbers of heads and relations, and with the Adam optimiser, all of which present qualitatively similar results.

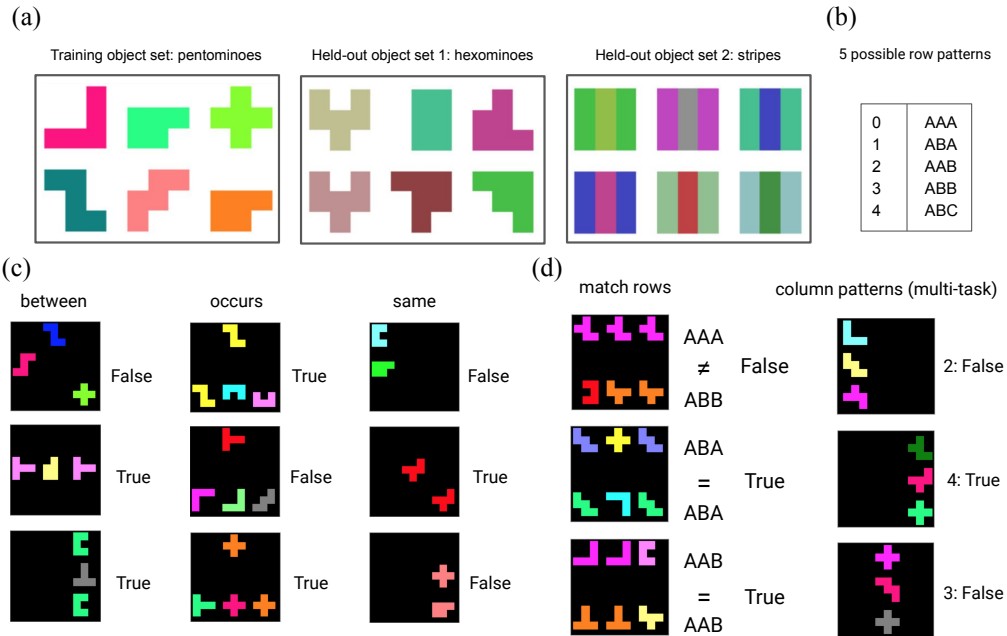

Figure 2: Relations Game object sets and tasks. (a) Example objects from the training set and held-out test sets. (b) There are five possible row / column patterns. In a multi-task setting, recognising each row pattern is a separate task. (c) Three examples tasks for the single-task setting. (d) An example target task (left) and curriculum (right) for the multi-task setting. The curriculum task ids (right) for each of the three examples (2, 4, and 3) correspond to the respective patterns in (b), and the task in each case is to confirm whether or not the column of objects in the image conform to the designated pattern. The aim of the target task (left) is to test whether the two rows of objects have the same pattern according to (b).

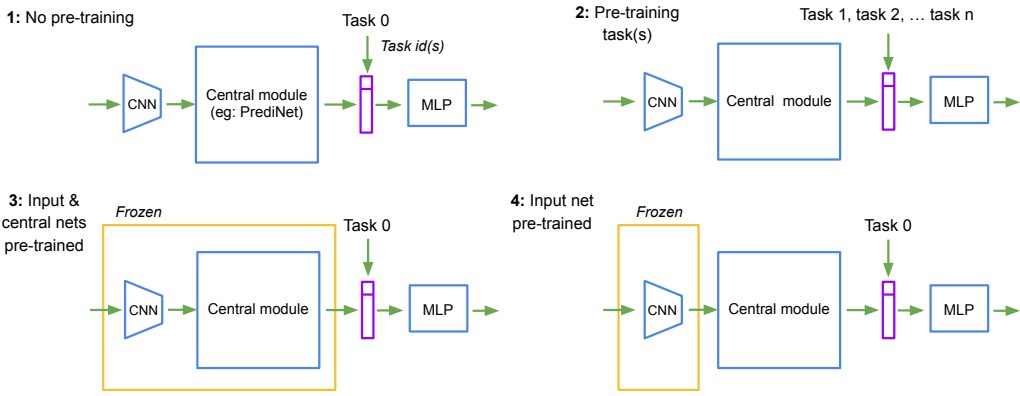

Figure 3: The four-stage experimental protocol for multi-task curriculum training. The same input module (CNN) and output module (MLP) are used for the PrediNet and all baseline architectures; only the central module varies. Task identifiers are appended to the central module's output vector.

Table 1: Data efficiency in a single-task Relations Game setting.

| Relation | Object set | MLP1 | MLP2 | RN | MHA | PrediNet |
|---|---|---|---|---|---|---|
| same | Hexominoes | 96.1±0.007 | 96.4±0.006 | 73.2±0.05 | 94.7±0.1 | **100**±0.0 |
| | Stripes | 93.3±0.01 | 94.0±0.01 | 72.9±0.05 | 93.7±0.1 | **100**±0.0 |
| between | Hexominoes | 98.7±0.005 | 98.8±0.004 | 70.8±0.01 | 89.2±0.1 | **99.2**±0.004 |
| | Stripes | 96.9±0.008 | 97.3±0.004 | 65.2±0.05 | 85.5±0.1 | **98.7**±0.007 |
| occurs | Hexominoes | 88.0±0.01 | 94.8±0.03 | 61.6±0.01 | 88.4±0.2 | **98.5**±0.009 |
| | Stripes | 73.2±0.03 | 87.3±0.07 | 62.6±0.02 | 80.8±0.1 | **96.9**±0.01 |
| xoccurs | Hexominoes | 81.5±0.02 | 84.4±0.04 | 55.0±0.009 | 54.7±0.008 | **95.4**±0.01 |
| | Stripes | 78.2±0.03 | 80.8±0.05 | 54.0±0.01 | 53.6±0.007 | **95.5**±0.01 |
| colour/shape | Hexominoes | 53.4±0.07 | 55.8±0.05 | 45.1±0.05 | 88.6±0.03 | **94.3**±0.01 |

To assess the generality and reusability of the representations produced by the PrediNet, we adopted a four-stage experimental protocol wherein 1) the network is pre-trained on a curriculum of one or more tasks, 2) the weights in the input CNN and PrediNet are frozen while the weights in the output MLP are re-initialised with random values, and 3) the network is retrained on a new target task or set of tasks (Fig. 3). In step 3, only the weights in the output MLP change, so the target task can only be learned to the extent that the PrediNet delivers re-usable representations to it, representations the PrediNet has learned to produce without exposure to the target task. To assess this, we can compare the learning curves for the target task with and without pre-training. We expect pre-training to improve data efficiency, so we should see accuracy increasing more quickly with pre-training than without it. For evidence of transfer, and to confirm the hypothesis of reusability, we are also interested in the final performance on the target task after pre-training, given that the weights of the pre-trained input CNN and PrediNet are frozen. This measure indicates how well a network has learned to form useful representations. The more different the target task is from the pre-training curriculum, the more impressed we should be that the network is able to learn the target task.

## 5 RESULTS

As a prelude to investigating the issues of generality and reusabilty, we studied the data efficiency of the PrediNet architecture in a single-task Relations Game setting. Results obtained on a selection of five tasks – 'same', 'between', 'occurs', 'xoccurs', and 'colour / shape' – are summarised in Table 1. The first three tasks are as described in Fig. 2. The 'xoccurs' relation is similar to occurs. It holds iff the object in the top row occurs in the bottom row and the other two objects in the bottom row are different. The 'colour / shape' task involves four labels, rather than the usual two: same-shape / same-colour; different-colour / same-shape; same-colour / different shape; different-colour / different shape. In the dataset for this task, each image contains two objects randomly placed, and one of the four labels must be assigned appropriately. Table 1 shows the accuracy obtained by each of the five architectures after 100,000 batches when tested on the two held-out object sets. The PrediNet is the only architecture that achieves over 90% accuracy on all tasks with both held-out object sets after 100,000 batches. On the 'xoccurs' task, the PrediNet out-performs the baselines by more than 10%, and on the 'colour / shape' task (where chance is 25%), it out-performs all the baselines except MHA by 25% or more.

Next, using the protocol outlined in Fig. 3, we compared the PrediNet's ability to learn re-usable representations with each of the baselines. We looked at a number of combinations of target tasks and pre-training curriculum tasks. Fig. 4 depicts our findings for one these combinations in detail, specifically three target tasks corresponding to three of the five possible column patterns (ABA, AAB, and ABB (Fig. 2d)), and a pre-training curriculum comprising the single 'between' task. The plots present learning curves for each of the five architectures at each of the four stages of the experimental protocol. In all cases, accuracy is shown for the 'stripes' held-out object set (not the training set). Of particular interest are the (green) curves corresponding to Stage 3 of the experimental protocol. These show how well each architecture learns the target task(s) after the central module has been

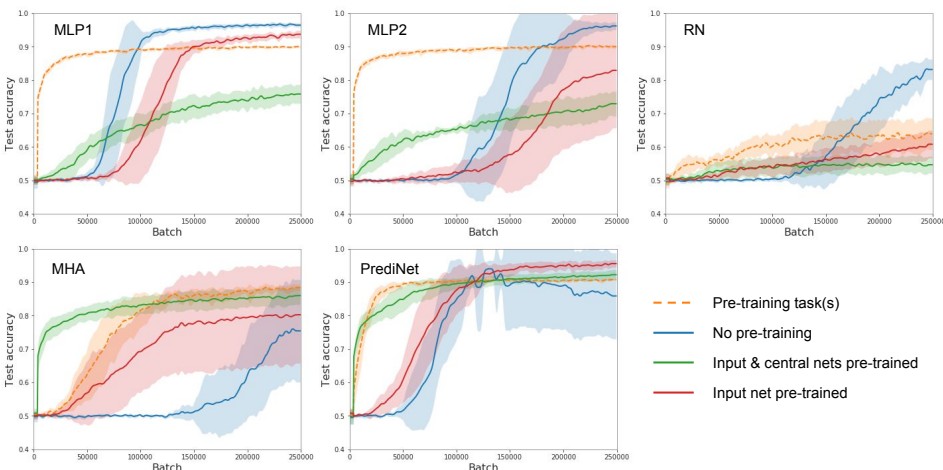

Figure 4: Multi-task curriculum training. The target tasks are three column patterns (AAB, ABA, and ABB) and the sole curriculum task is the 'between' relation.

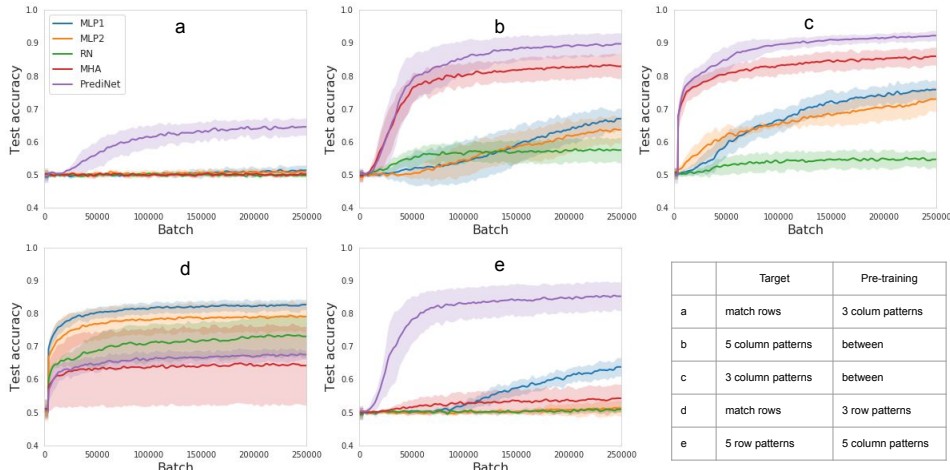

Figure 5: Reusability of representations learned with a variety of target and pre-training tasks.

pre-trained on the curriculum task(s) and its weights are frozen. The PrediNet learns faster than any of the baselines, and is the only one to achieve an accuracy of 90%. The rapid reusability of the representations learned by both the MHA baseline and the PrediNet is noteworthy because the 'between' relation by itself seems an unpromising curriculum for subsequently learning the AAB and ABB column patterns. As the (red) curve for Stage 4 of the protocol shows, the reusability of the PrediNet's representations cannot be accounted for by the pre-training of the input CNN alone.

Fig. 5 shows a larger range of target task / curriculum task combinations, concentrating exclusively on the Stage 3 learning curves. Here a more complete picture emerges. In both Fig. 5a and Fig. 5d the target task is 'match rows' (Fig. 2d), but they differ in their pre-training curricula. The curriculum for Fig. 5d is three of the five row patterns (ABA, AAB, and ABB). This is the only case where the PrediNet does not learn representations that are more useful for the target task than those of all the baselines, outperforming only two of the four. However, when the curriculum is the three analogous *column* patterns rather than row patterns, the performance of all four baselines collapses to chance, while the PrediNet does well, attaining similar performance as for the row-based curriculum (Fig. 5a). This suggests the PrediNet is able to learn representations that are orientation invariant, which aids transfer. This hypothesis is supported by Fig. 5e, where the target tasks are all five row patterns, while the curriculum is all five column patterns. None of the baselines is able to learn reusable representations in this context; all remain at chance, whereas the PrediNet achieves 85% accuracy.

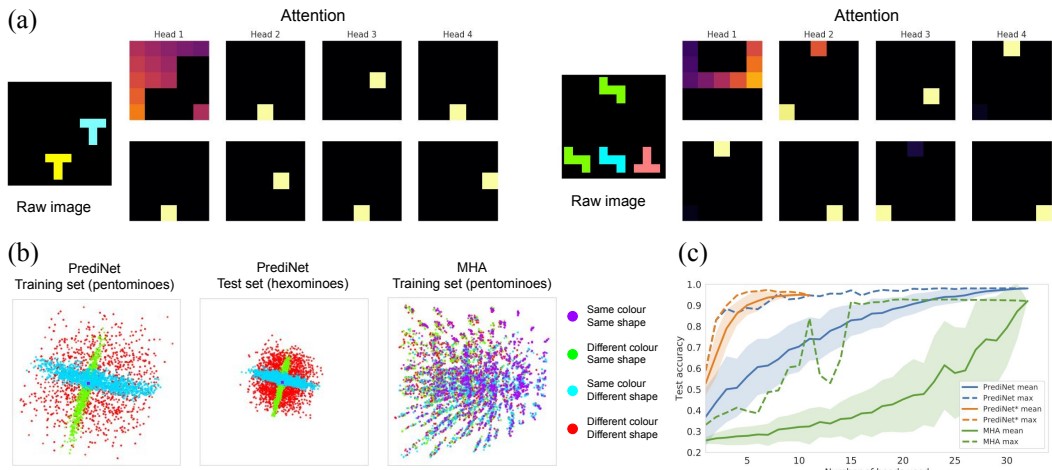

Figure 6: (a) Attention heat maps for the first four heads of a trained PrediNet. Left: trained on the 'same' task. Right: trained on the 'occurs' task. (b) Principal component analysis. Left: PCA on the output of a selected head for a PrediNet trained on the 'colour / shape' task for pentominoes images (training set). Centre: The same PrediNet applied to hexominoes (held-out test set). Right: PCA applied to a representative head of the MHA baseline with pentominoes (training set). (c) Ablation study. Accuracy for PrediNet and MHA on the 'colour / shape' task when random subsets of the heads are used at test time. PrediNet* only samples from heads that attend to the two objects.

To better understand the operation of the PrediNet, we carried out a number of visualisations. One way to find out what the PrediNet's heads learn to attend is to submit images to a trained network and, for each head $h$, apply the two attention masks $\mathrm{softmax}(Q_1^h K^\top)$ and $\mathrm{softmax}(Q_2^h K^\top)$ to each of the $n$ feature vectors in the convolved image $L$. The resulting matrix can then be plotted as a heat map to show how attention is distrubuted over the image. We did this for a number of networks trained in the single-task setting. Fig. 6a shows two examples, and the Supplementary Material contains a more extensive selection. As we might expect, most of the attention focuses on the centres of single objects, and many of the heads pick out pairs of distinct objects in various combinations. But some heads attend to halves or corners of objects. Although most attention is focal, whether directed at object centres or object parts, some heads exhibit diffuse attention, which is possible thanks to the soft key-query matching mechanism. So the PrediNet can (but isn't forced to) treat the background as a single entity, or to treat an identical pair of objects as a single entity.

To gain some insight into how the PrediNet encodes relations, we carried out principal component analysis (PCA) on each head of the central module's output vectors for a number of trained networks, again in the single-task setting (Fig. 6b). We chose the four-label 'colour / shape' task to train on, and mapped 10,000 example images onto the first two principal components, colouring each with their ground-truth label. We found that, for some heads, differences in colour and shape appear to align along separate axes (Fig. 6b). This contrasts with the MHA baseline, whose heads don't seem to individually cluster the labels in a meaningful way. For the other baselines, which lack the multi-head organisation of the PrediNet and the MHA network, the only option is to carry out PCA on the whole output vector of the central module. Doing this, however, does not produce interpretable results for any of the architectures (Fig.S8). We also identified the heads in the PrediNet that attended to both objects in the image and found that they overlapped almost entirely with those that meaningfully clustered the labels (Fig.S10). Finally, still using the 'shape / colour task', we carried out an ablation study, which showed that the PrediNet is significantly more robust than the MHA network to pruning a random subset of heads at test time. Moreover, if pruned to leave only those heads that attended to the two objects, the performance of the full network could be captured with just a handful of heads (Fig. 6c). Taken together, these results are suggestive of something we might term *relational disentangling* in the PrediNet.

Finally, to flesh out the claim that the PrediNet generates explicitly relational representations according to the semantics of Equation 1, we extended the PrediNet module to generate an additional output in the form of a Prolog program (Fig. 7). This involves assigning symbolic identifiers 1) to each of the

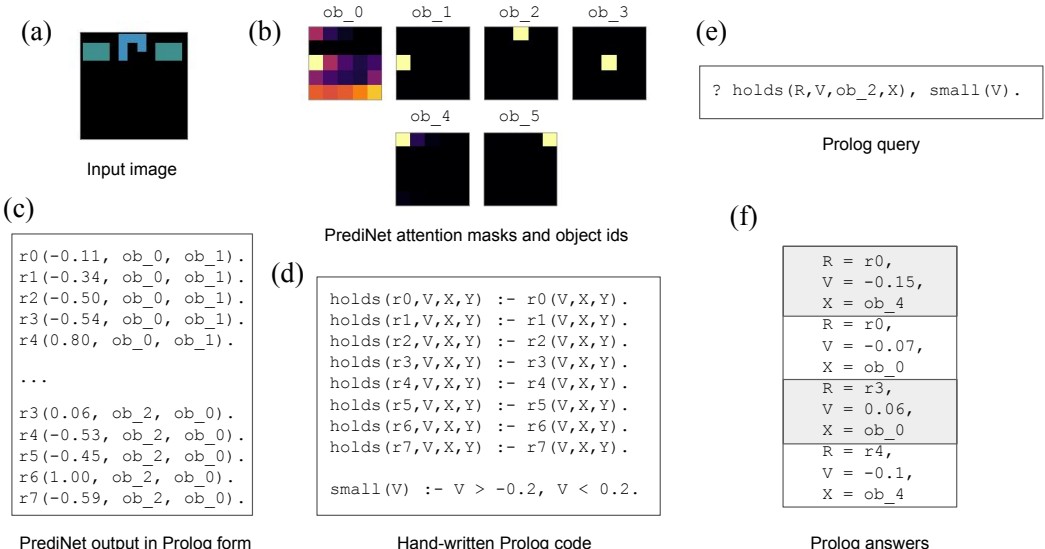

Figure 7: PrediNet output in propositional form. (a) A small PrediNet (8 heads, 8 relations) trained on the 'between' task is given an image. (b) Mean shift clustering is applied to the set of all attention masks computed by the heads. Each of the resulting 6 clusters is assigned a symbolic identifier. (c) Each relation is also given a symbolic identifier, and all 64 propositions computed by the PrediNet are enumerated in Prolog syntax, in accordance with Equation 1. (A subset is shown.) (d) The results can be combined with further hand-written Prolog clauses. (Upper-case letters denote variables, while constants start with lower-case letters.) (e) Prolog queries can then be submitted. Here we are asking which relations $r$ hold with a small value $v$ between ob_2 and any other object $x$. (f) The query yields four answers.

PrediNet's $j$ relations, and 2) to every object picked out by its $k$ heads via the attention masks they compute. Then the corresponding $j \times k$ propositions can be enumerated in Prolog syntax. Assigning symbolic identifiers to the relations is trivial. But because attention masks can differ slightly even when they ostensibly pick out the same region of the input image, it's necessary to cluster them before assigning symbolic identifiers to the corresponding objects. We used mean shift clustering for this. Fig. 7 presents a sample of the PrediNet's output in Prolog form, along with an example of deductive inference carried out with this program. The example shown is not intended to be especially meaningful; without further analysis, we lack any intuitive understanding of the relations the PrediNet has discovered. But it demonstrates that the representations the PrediNet produces can be understood in predicate calculus terms, and that symbolic deductive inference is one way (though not the only way) in which they might be deployed downstream.

## 6 RELATED WORK

The need for good representations has long been recognised in AI (McCarthy, 1987; Russell & Norvig, 2009), and is fundamental to deep learning (Bengio et al., 2013). The importance of reusability and abstraction, especially in the context of transfer, is emphasised by Bengio, *et al.* (Bengio et al., 2013), who argue for feature sets that are "invariant to the irrelevant features and disentangle the relevant features". Our work here shares this motivation. Other work has looked at learning representations that are disentangled at the feature level (Higgins et al., 2017a; 2018). The novelty of the PrediNet is to incorporate architectural priors that favour representations that are disentangled at the relational and propositional levels. Previous work with relation nets and multi-head attention nets has shown how non-local information can be extracted from raw pixel data and used to solve tasks that require relational reasoning. (Santoro et al., 2017; Palm et al., 2018; Santoro et al., 2018; Zambaldi et al., 2019) But unlike the PrediNet, these networks don't produce representations with an explicitly relational, propositional structure. By addressing the problem of acquiring structured representations, the PrediNet complements another thread of related work, which is concerned with learning how to

carry out inference with structured representations, but which assumes the job of acquiring those representations is done elsewhere (Getoor & Taskar, 2007; Battaglia et al., 2016; Rocktäschel & Riedel, 2017; Evans & Grefenstette, 2018; Dong et al., 2019).

In part, the present work is motivated by the conviction that curricula will be essential to lifelong, continual learning in a future generation of RL agents if they are to exhibit more general intelligence, just as they are for human children. Curricular pre-training has a decade-long pedigree in deep learning (Bengio et al., 2009). Closely related to curriculum learning is the topic of transfer (Bengio, 2012), a hallmark of general intelligence and the subject of much recent attention (Higgins et al., 2017b; Kansky et al., 2017; Schwarz et al., 2018). The PrediNet exemplifies a different (though not incompatible) viewpoint on curriculum learning and transfer from that usually found in the neural network literature. Rather than (or as well as) a means to guide the network, step by step, into a favourable portion of weight space, curriculum learning is here viewed in terms of the incremental accumulation of propositional knowledge. This necessitates the development of a different style of architecture, one that supports the acquisition of propositional, relational representations, which also naturally subserve transfer.

Asai, whose paper was published while the present work was in progress, describes an architecture with some similarities to the PrediNet, but also some notable differences (Asai, 2019). For example, Asai's architecture assumes an input representation in symbolic form where the objects have already been segmented. By contrast, in the present architecture, the input CNN and the PrediNet's dot-product attention mechanism together learn what constitutes an object.

## 7 CONCLUSION AND FURTHER WORK

We have presented a neural network architecture capable, in principle, of supporting predicate logic's powers of abstraction without compromising the ideal of end-to-end learning, where the network itself discovers objects and relations in the raw data and thus avoids the symbol grounding problem entailed by symbolic AI's practice of hand-crafting representations (Harnad, 1990). Our empirical results support the view that a network architecturally constrained to learn explicitly propositional, relational representations will have beneficial data efficiency, generalisation, and transfer properties. Although, the present experiments don't use the fully propositional version of the PrediNet output, the concatenated vector form inherits many of its beneficial properties, notably a degree of compositionality. In particular, one important respect in which the PrediNet differs from other network architectures is the extent to which it *canalises* information flow; at the core of the network, information is organised into small chunks which are processed in parallel channels that limit the ways the chunks can interact. We believe this pressures the network to learn representations where each separate chunk of information (such as a single value in the vector $R*$) has independent meaning and utility. (We see evidence of this in the relational disentanglement of Fig. 6.) The result is a representation whose component parts are amenable to recombination, and therefore re-use in a novel task. But the findings reported here are just the first foray into unexplored architectural territory, and much work needs to be done to gauge the architecture's full potential.

The focus of the present paper is the *acquisition* of propositional representations rather than their use. But thanks to the structural priors of its architecture, representations generated by a PrediNet module have a natural semantics compatible with predicate calculus (Equation 1), which makes them an ideal medium for logic-like downstream processes such as rule-based deduction, causal or counterfactual reasoning, and inference to the best explanation (abduction). One approach here would be to stack PrediNet modules and / or make them recurrent, enabling them to carry out the sort of iterated, sequential computations required for such processes (Palm et al., 2018; Dehghani et al., 2019). Another worthwhile direction for further research would be to develop reinforcement learning (RL) agents using the PrediNet architecture. One form of inference of particular interest in this context is model-based prediction, which can be used to endow an RL agent with look-ahead and planning abilities (Racanière et al., 2017; Zambaldi et al., 2019). Our expectation is that RL agents in which explicitly propositional, relational representations underpin these capacities will manifest more of the beneficial data efficiency, generalisation, and transfer properties suggested by the present results. As a stepping stone to such RL agents, the Relations Game family of datasets could be extended into the temporal domain, and multi-task curricula developed to encourage the acquisition of temporal, as well as spatial, abstractions.

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

Table S2: Default hyperparameters

| Parameter | Value |
|---|---|
| Input images size | $36 \times 36 \times 3$ |
| $L$ size | $25 \times 34$ |
| Runs per experiment | 10 |
| Optimiser | Gradient descent |
| Learning rate | 0.01 |
| Batch size | 10 |
| Input CNN output channels | 32 |
| Input CNN filter size | 12 |
| Input CNN stride | 6 |
| Input CNN activation | ReLu |
| Bias | Yes |
| Output MLP hidden layer size | 8 |
| Output MLP output size | 2 (4) |
| Output MLP activation | ReLu |
| Bias | Yes (both) |
| MLP1 output size | $k(j + 4)$ |
| MLP1 activation | ReLu |
| Bias | Yes |
| MLP2 hidden layer size | 1024 |
| MLP2 activations | ReLu |
| MLP2 output size | $k(j + 4)$ |
| Bias | Yes (both) |
| RN MLP hidden layer size (pre-aggregation) | 256 |
| RN output size | $k(j + 4)$ |
| RN activation | ReLu |
| RN aggregation | Element-wise mean |
| Bias | No |
| MHA no. of heads | $k$ |
| MHA key / query size | 16 |
| MHA value size | $j + 4$ |
| MHA output size | $k(j + 4)$ |
| MHA attention mechanism | $\text{softmax}(QK^\top)V$ |
| Bias | No |
| PrediNet no. of heads | $k = 32$ |
| PrediNet key / query size | $g = 16$ |
| PrediNet relations | $j = 16$ |
| PrediNet output size | $k(j + 4)$ |
| Bias | n/a |

## S1 HYPERPARAMETERS

Table S2 shows the default hyperparameters used for the experiments reported in the main text.

## S2 SUPPLEMENTARY ANALYSIS

### S2.1 DIMENSIONALITY REDUCTION ON INTERMEDIATE REPRESENTATIONS

To qualitatively assess the nature of the representations produced by each architecture, we performed a dimensionality reduction analysis on the outputs of the central module of each architecture trained on the 'colour / shape' task. After training, a batch of 10,000 images (pentominoes) was passed through

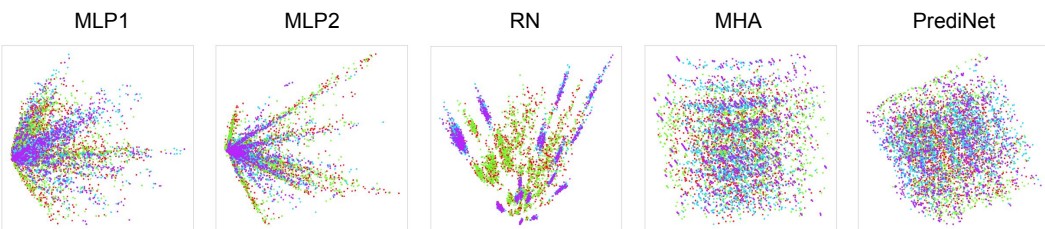

Figure S8: Representative central module outputs for networks trained on the 'colour / shape' task when projected onto the two largest principal components.

the network and principal component analysis (PCA) was performed on the resulting representations, which were then projected onto the two largest principal components for visualisation. The projected representations were then colour-coded by the labels for the corresponding images (i.e. different/same shape, different/same colour).

PCA on the full representations (concatenating the head outputs in the case of the PrediNet and MHA models) did not yield any clear clustering of representations according to the labels for any of the models (Figure S8).

For the PrediNet and MHA models, we also ran separate PCAs on the output relations of each head in order to see how distributed / disentangled the representations were. While in the MHA model there was no evidence of clustering by label on any of the heads, reflecting a heavily distributed representation, there were several heads in the PrediNet architecture that individually clustered the different labels (Figure S9). In some heads, colour and shape seemed to be projected along separate axes (e.g. heads 5, 26, and 27), while in others objects with different colours seemed to be organised in a hexagonal grid (e.g. heads 9 and 14).

We noted that the clustering was preserved (though slightly compressed in PC space) when the held-out set of images (hexominoes) was passed through the PrediNet and projected onto the same principal components derived using the training set. (In Section S2.2, we show that the PrediNet heads that seem to cluster the labels also attend to the two objects in the image rather than the background.)

## S2.2 ATTENTION ANALYSIS

To assess the extent to which the various PrediNet heads attend to actual objects as opposed to the background, we produced a lower resolution *content* mask for each image (with the same resolution as the attention mask) containing $0.0$s at locations where there are no objects in the corresponding pixels of the full image, $1.0$s where more than $90\%$ of the pixels contain an object, and $0.5$s otherwise. By applying the attention mask to the content mask, and summing the resulting elements, we produced a scalar indicating whether the attention mask was selecting a region of the image with an object (value close to $1.0$), or the background (value close to $0.0$). This was tested over $1000$ images from the training set (Fig. S10), but similar results are obtained if the held-out set images are used instead. The top plot in Fig. S10 shows that both attention masks of some heads consistently attend to objects, while others to a combination of object and background. Importantly, the heads for which the PCA meaningfully clusters the labels are also the ones in which both attention masks attend to objects (Fig. S9).

We additionally provide a similar analysis with a *position* mask, where each pixel in the mask contains a unique location index. The middle plot in Fig. S10 shows that the attention masks in the majority of the heads do not consistently attend to specific locations. Finally, the mean absolute values of the per-head input weights to the output MLP are shown in the bottom plot of the same figure. Interestingly, the heads that consistently attend only to objects have higher weighting than the rest.

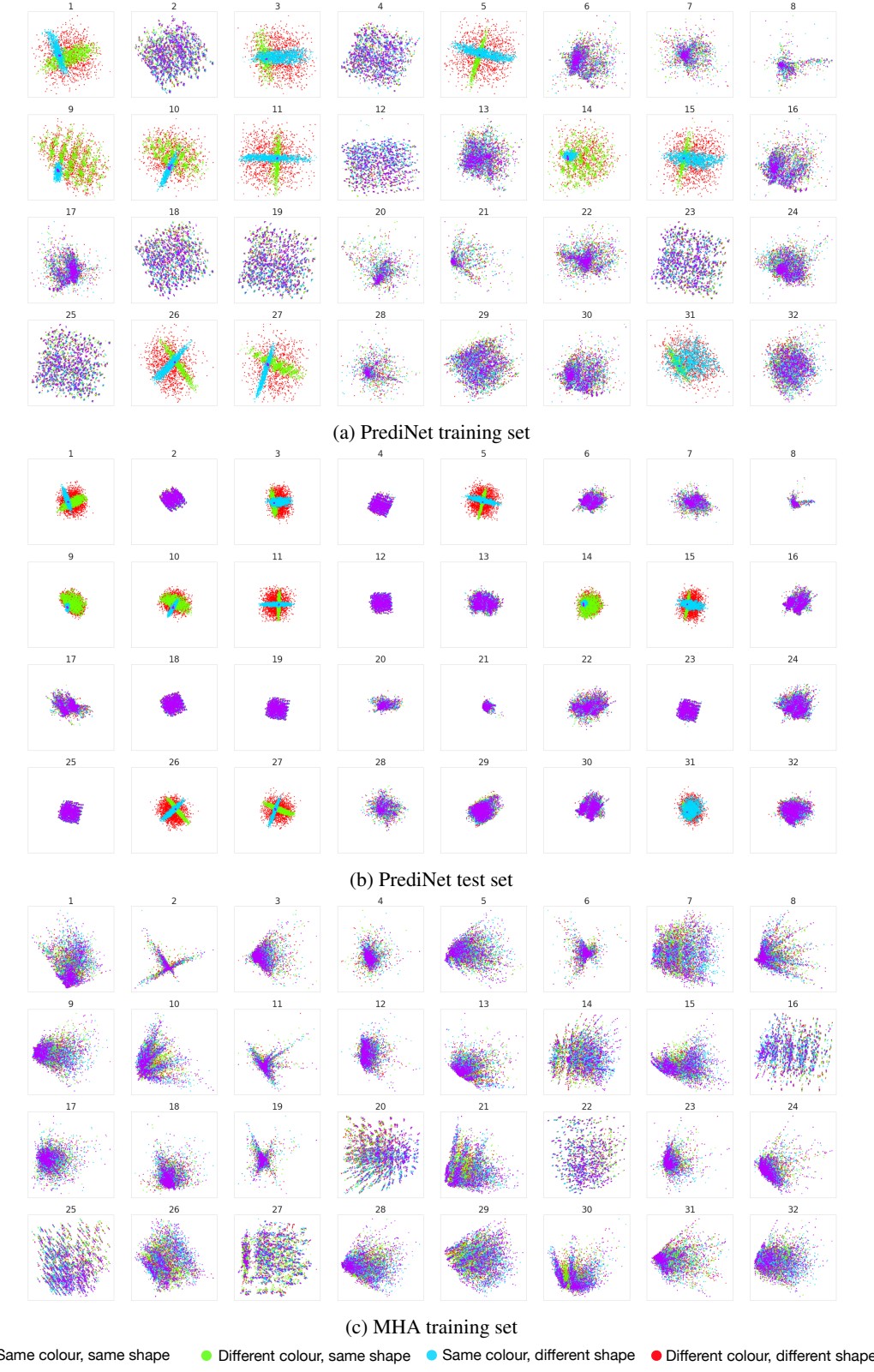

Figure S9: Per-head PCA on the heads of a PrediNet and an MHA trained on the 'colour / shape' task. For all networks, PCA was performed using the training data (pentominoes). In (a) and (c), the training data are projected onto the two largest PCs and in (b) the test data (hexominoes) was used.

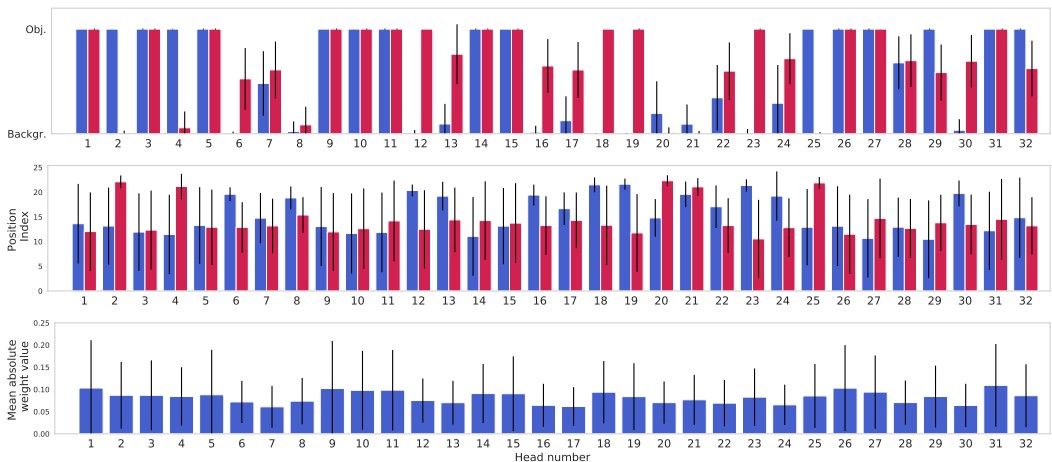

Figure S10: Top: The extent to which the two attention masks of the different heads attend to objects rather than the background. Middle: The extent to which the two attention masks of the different heads attend to specific locations in the image. Bottom: Mean absolute value of the weights from the different PrediNet heads to the output MLP.

## S3  EXPERIMENTAL VARIATIONS

Further experimental results are provided in this section, including variations in hyper-parameters. Fig. S11 presents test accuracy curves for the 'stripes' object set, for which a summary is presented in Table 1 of the main text. Fig. S12 shows the results on the same experiment but using the Adam optimiser instead of SGD, with a learning rate of $10^{-4}$. The TensorFlow default values for all other Adam parameters were used. While other learning rate values were also tested, a value of $10^{-4}$ gave the best overall performance for all architectures. Multi-task experiments were also performed using Adam with the same learning rate (Fig. S14 & S15), yielding an overall similar performance to SGD with a learning rate of $10^{-2}$.

To assess the extent to which the number of heads and relations plays a role in the performance, we ran experiments with $k = 64$ heads and $j = 16$ relations (Fig. S16 & S17), as well as $k = 16$ heads and $j = 32$ relations (Fig. S18 & S19). The results indicate that having a greater number of heads leads to better performance than having a greater number of relations, because they provide more stability during training and, perhaps, a richer propositional representation.

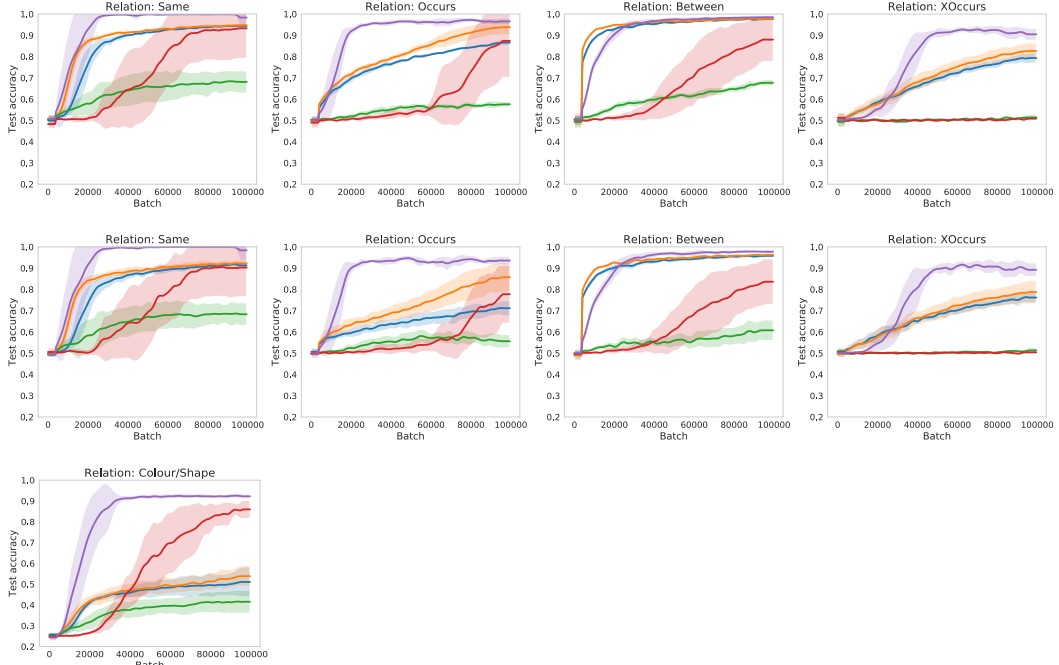

Figure S11: Relations Game learning curves for the different models. SGD with a learning rate of 0.01 was used with a PrediNet of $k = 32$ heads and $j = 16$ relations. The top and bottom rows show results for the 'hexominoes' held-out object set, while the middle row is for the 'stripes' held-out object set. The results for the top 10 batches are summarised in Table 1 of the main manuscript.

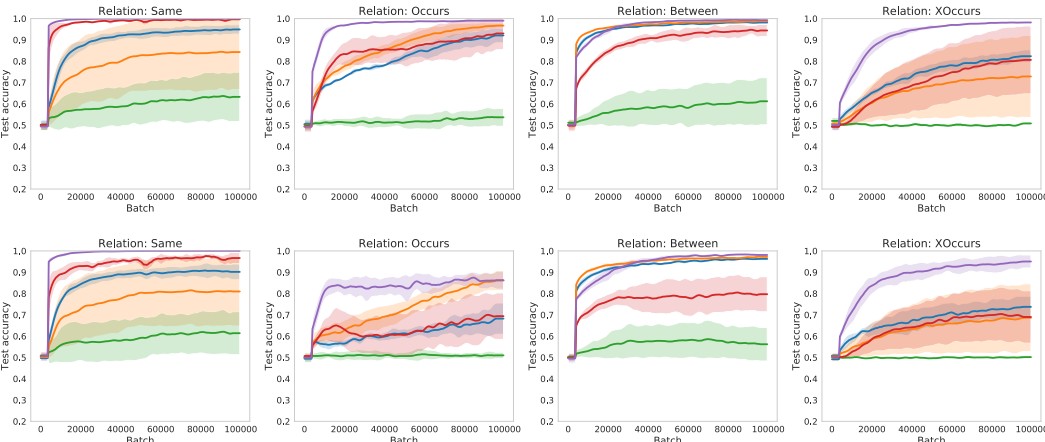

Figure S12: Relations Game learning curves for the different models trained with the Adam optimiser (learning rate: $10^{-4}$). All other experimental parameters are the same as Fig. S11. The top row shows results for the 'hexominoes' held-out object set, while the bottom row is for the 'stripes' held-out object set

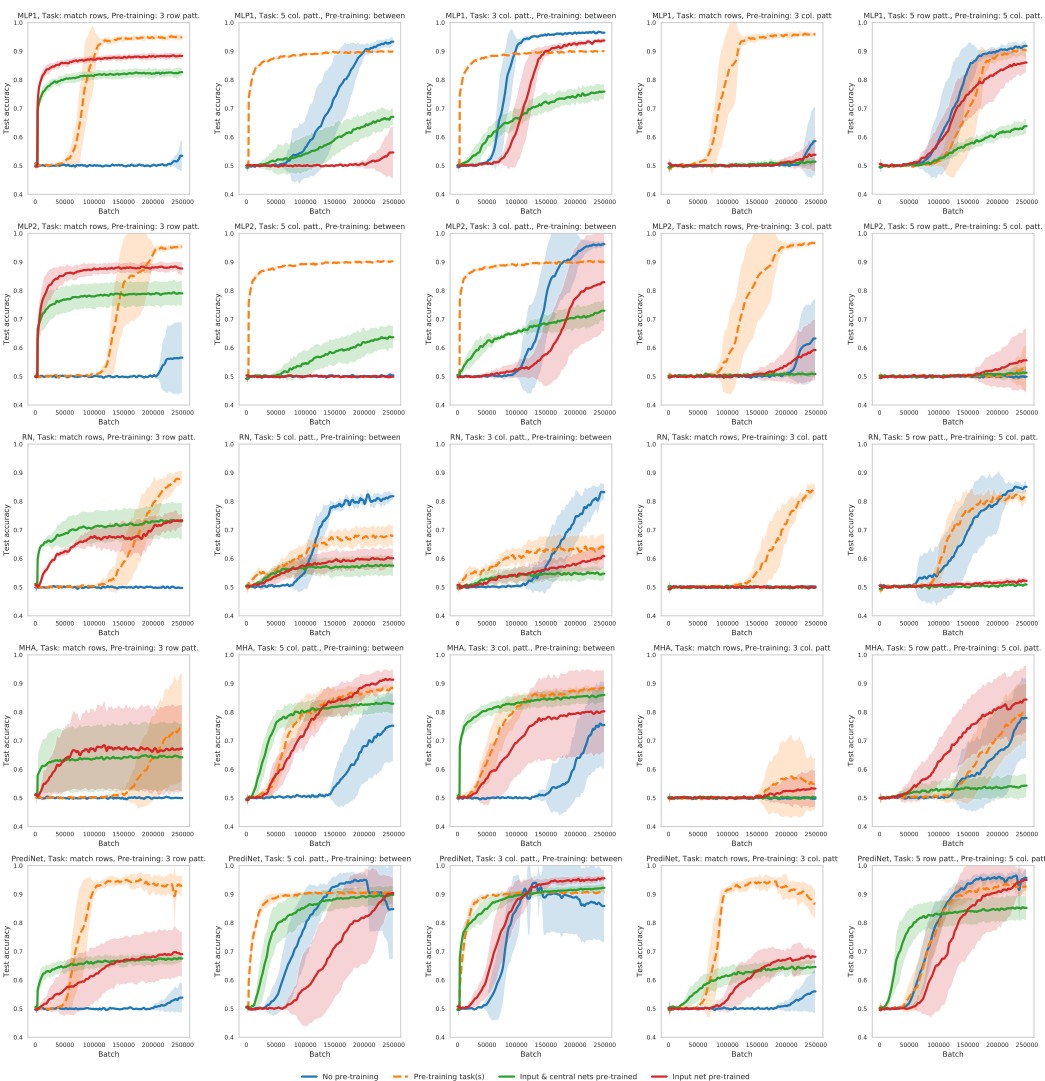

Figure S13: Multi-task curriculum training. The columns correspond to different target / pre-training task combinations, while the rows correspond to the different architectures. SGD with a learning rate of 0.01 was used, with $k = 32$ and $j = 16$. Training was performed using the pentominoes object set and testing using the 'stripes' object set. From left to right, the combinations of target / pre-training tasks are: ('match rows', '3 row patterns'), ('5 column patterns', 'between'), ('3 column patterns', 'between'), ('match rows', '3 column patterns') and ('5 row patterns', '5 column patterns'). From top to bottom, the different architectures are: MLP1, MLP2, relation net (RN), multi-head attention (MHA) and PrediNet.

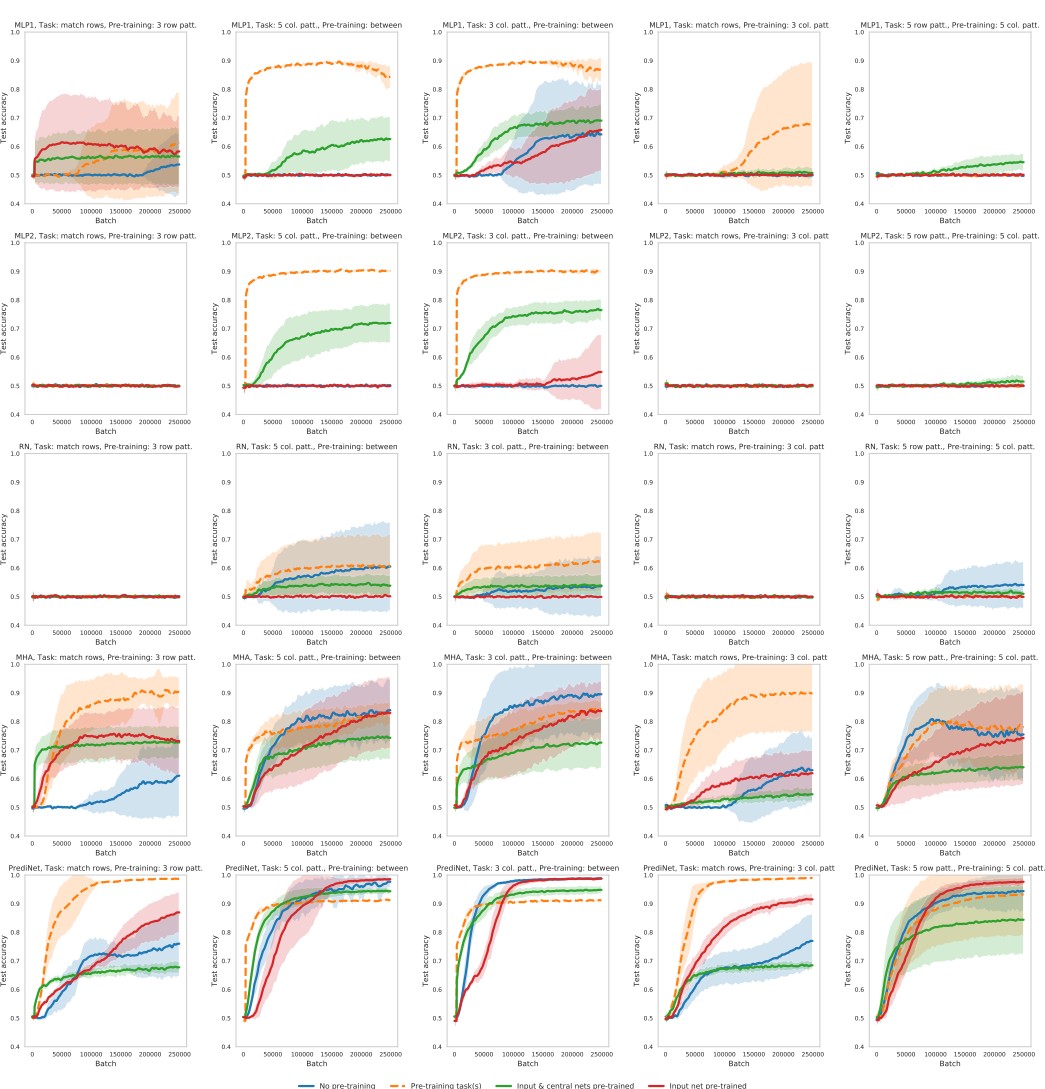

Figure S14: Multi-task curriculum training. The columns correspond to different target / pre-training task combinations, while the rows correspond to the different architectures, as in Fig. S13. The Adam optimiser with a learning rate of $10^{-4}$ was used. Training was performed using the pentominoes object set and testing using the 'stripes' object set.

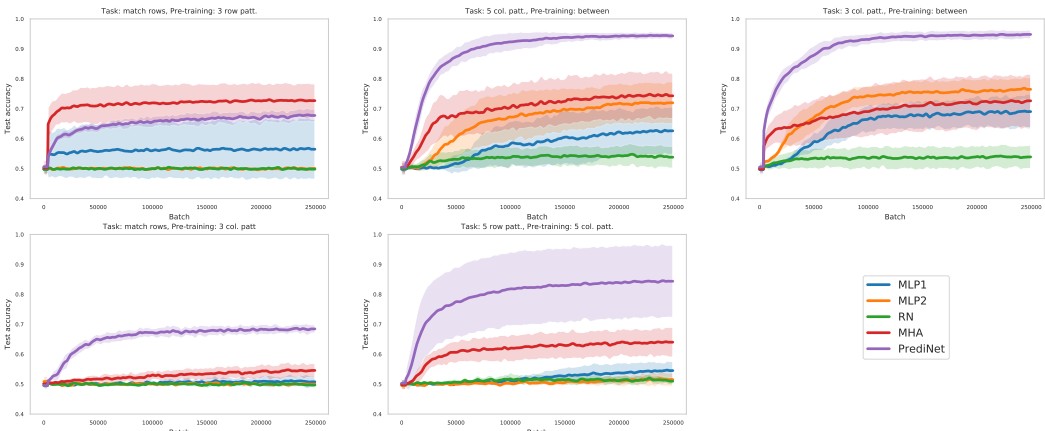

Figure S15: Reusability of representations learned with a variety of target and pre-training tasks, using the 'stripes' object set. All architectures were trained using Adam, with a learning rate of $10^{-4}$. The experimental setup is the same as in Fig. S14.

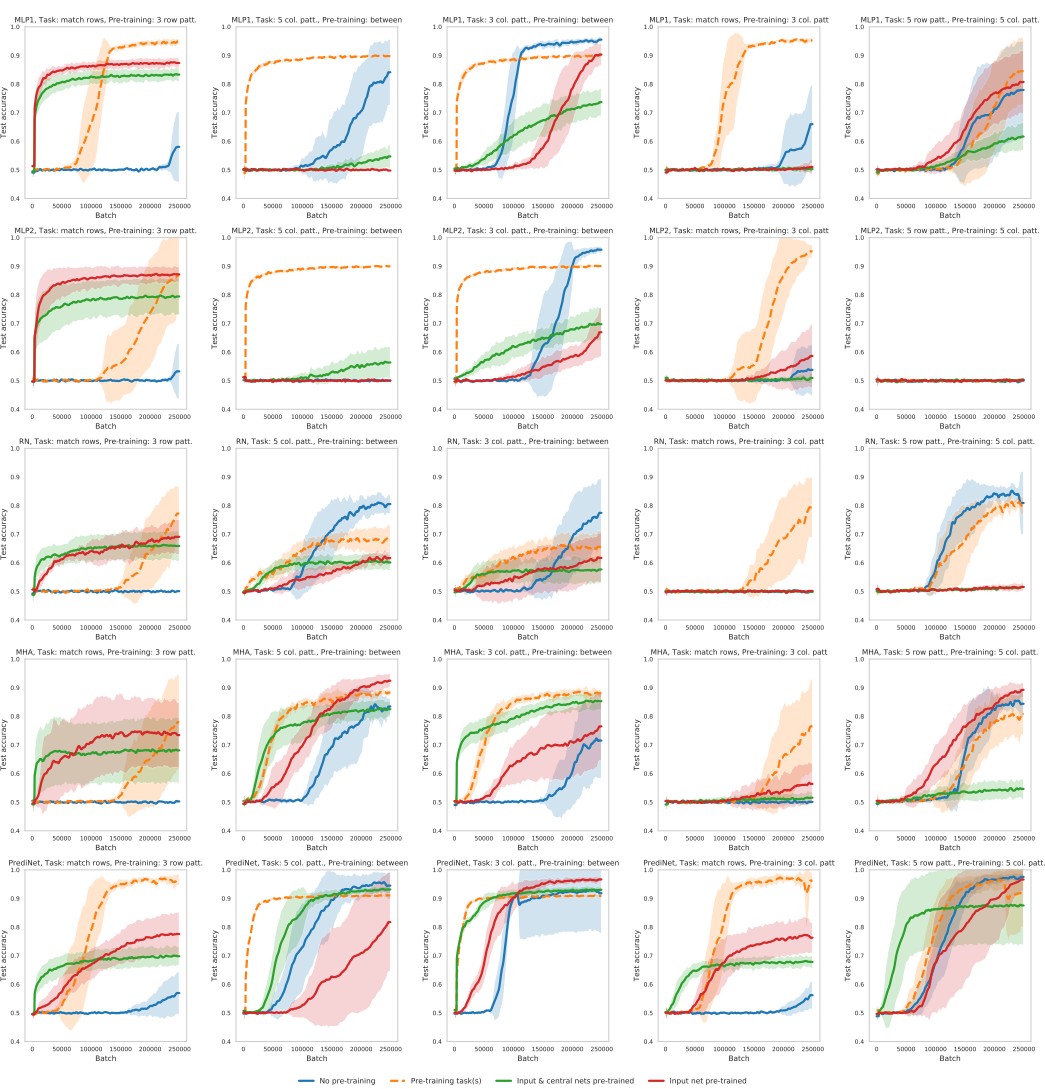

Figure S16: Multi-task curriculum training. The columns correspond to different target/pre-training task combinations, while the rows correspond to the different architectures. SGD with a learning rate of $0.01$ was used. Training was performed using the pentominoes object set and testing using the 'stripes' object set. The experimental setup is the same as for Fig. S13, except that $k = 64$ and $j = 16$. Increasing the number of heads for the PrediNet increases the stability during training and overall performance.

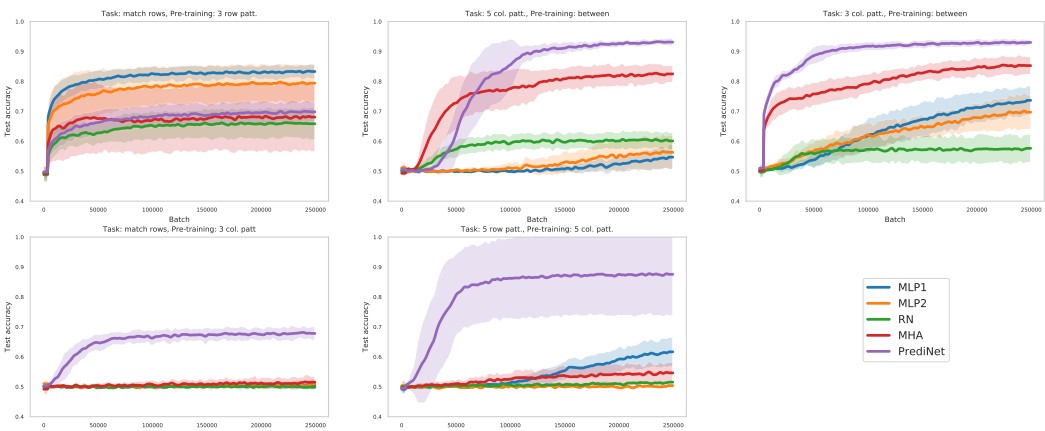

Figure S17: Reusability of representations learned with a variety of target and pre-training tasks, using the 'stripes' object set. All experimental parameters are as in Fig. S16.

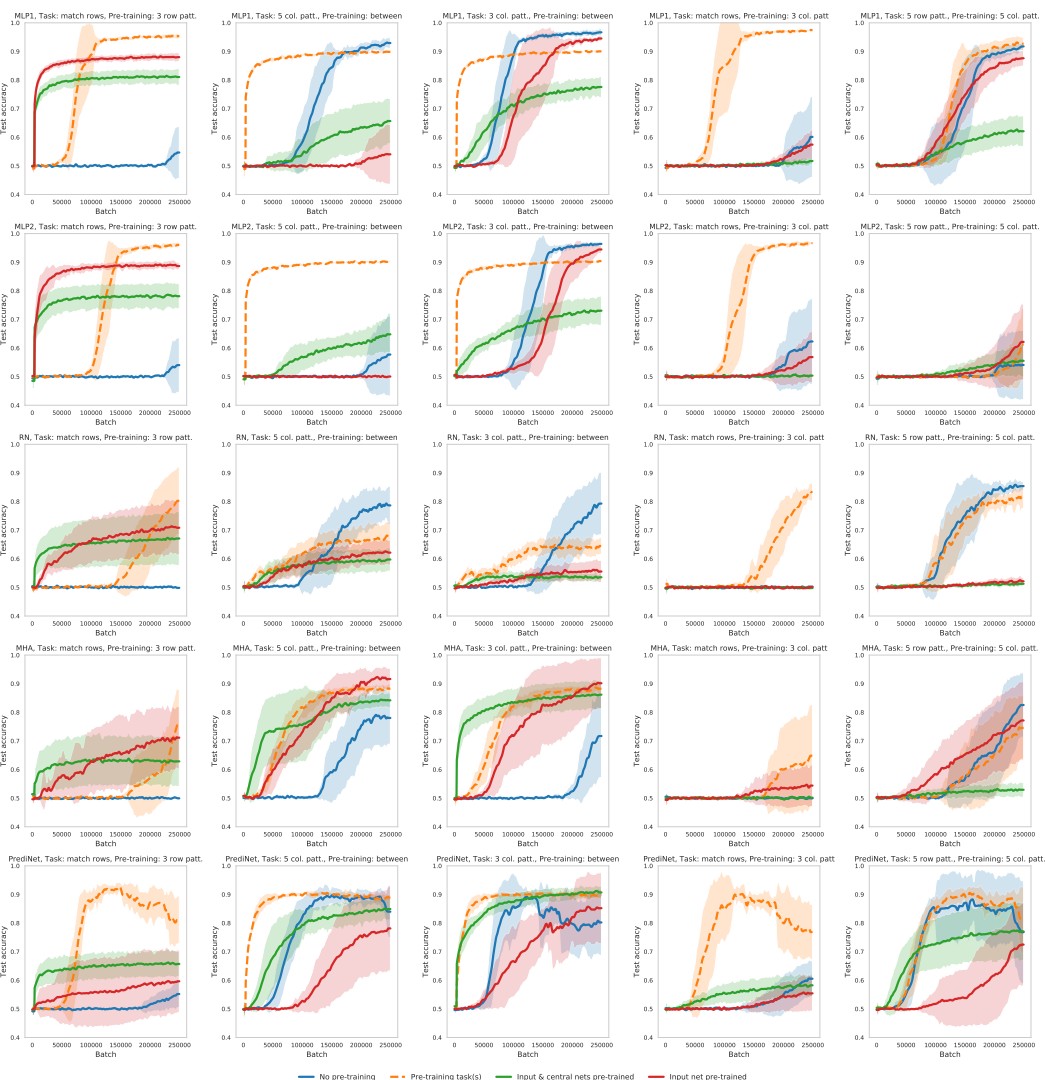

Figure S18: Multi-task curriculum training. The columns correspond to different target / pre-training task combinations, while the rows correspond to the different architectures. SGD with a learning rate of 0.01 was used. Training was performed using the pentominoes object set and testing using the 'stripes' object set. The experimental setup is the same as for Fig. S13, except that $k = 16$ and $j = 32$. Having fewer heads leads to a decrease in performance, even if the number of relations is increased to maintain network size.

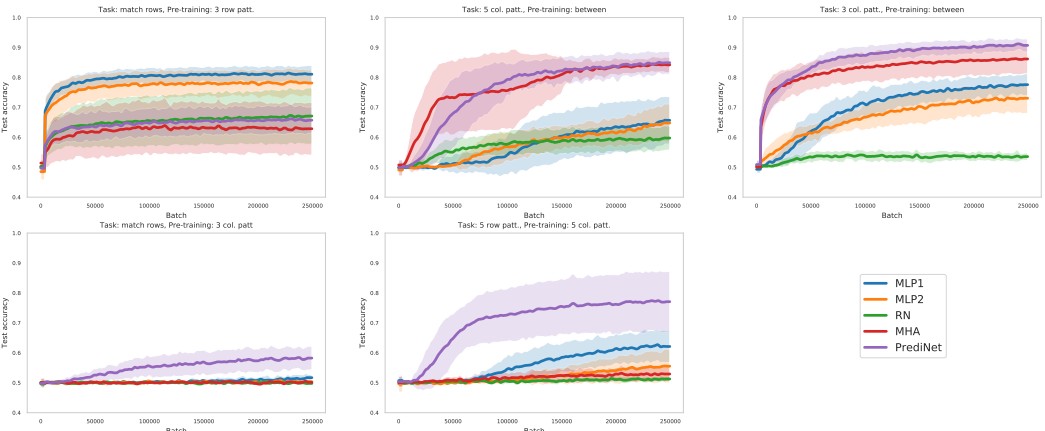

Figure S19: Reusability of representations learned with a variety of target and pre-training tasks, using the 'stripes' object set. All experimental parameters are as in Fig. S18.

