# OpenReview forum: "An Explicitly Relational Neural Network Architecture"
_ICLR.cc/2020/Conference — Reject_

### Official Review · AnonReviewer2 · 2019-10-23
**Official Blind Review #2**

**Rating:** 6

**Review:**

This paper presents a network architecture based on the multi-head self-attention module to learn a new form of relational representations. The proposed method is shown to improve data efficiency and generalization ability on a sequence of curriculum learning tasks.

+ The major novelty of this paper is a new attention-based network architecture that aims to discover objects and the relations between them. The idea of explicitly appending the patch positions to the representations is interesting, though I am not sure whether it can be generalized to real data.

- Since the proposed network takes patches of full images as inputs, my major concern is about its effectiveness on high-dimensional images with more realistic objects as in the CLEVR dataset other than 2d-grid objects. It would be better if the authors could extend their method to natural images.

- A closely related work is the NLM model [1], which can perfectly generalized to new tasks. Please compare the proposed model with it.

- Minor: Figures are understandable, but some of them are too small, especially for the graphical legend. As space is an issue, I would suggest removing some plots and increasing the size of the ones provided.

- In Figure 1, is g equal to n?

[1] Neural Logic Machines. Dong et al., ICLR 2019.

**Experience Assessment:**

I have read many papers in this area.

**Review Assessment: Checking Correctness Of Derivations And Theory:**

N/A

**Review Assessment: Checking Correctness Of Experiments:**

I assessed the sensibility of the experiments.

**Review Assessment: Thoroughness In Paper Reading:**

I read the paper at least twice and used my best judgement in assessing the paper.

---

> ### Author Response · Authors · 2019-11-07
> **Response to Reviewer #2**
>
> Many thanks for the review and suggestions. We’ll respond to them in reverse order. (If you feel we’ve dealt with them well, please consider increasing your score.)
>
>
> Reviewer #2: “In Figure 1, is g equal to n?”
>
> Response: They aren’t the same. n is the number of feature vectors output by the CNN. g is the size of the keys and queries. (If you feel this isn’t clear from Fig. 1 we could try to squeeze in a sentence stating this explicitly, but we are very pressed for space.)
>
>
> Reviewer #2: “I would suggest removing some plots and increasing the size of the ones provided.”
>
> Response:
>
> Sorry the text is so small in the figures. We hope it didn’t make the paper too hard to read. But the resolution is high, so everything should be legible if viewed on screen. As you point out, space is an issue; we’re now at the very limit. However, we’re extremely reluctant to relegate any more plots to the Supplementary Material. So we hope you will let us off the hook :-)
>
>
> Reviewer #2: “A closely related work is the NLM model [1], which can perfectly generalized to new tasks. Please compare the proposed model with it.”
>
> Response:
>
> Thank you for drawing our attention to this work (Dong et al (2019)). We have added it to the bibliography of the revised paper, and cited it in the Related Work section. As we point out there, Dong et al tackle an orthogonal problem to the one we’ve tackled. Dong et al’s NLM carries out inductive inference, assuming input is given in terms of propositions, relations, and objects. Our primary contribution is an architecture (the PrediNet) that _discovers_ objects and relations in the raw data, and learns to generate representations in propositional form built out of the objects and relations it has discovered. So, like the other papers we cite in this context, their work complements ours.
>
>
> Reviewer #2: “my major concern is about its effectiveness on high-dimensional images with more realistic objects as in the CLEVR dataset other than 2d-grid objects. It would be better if the authors could extend their method to natural images.”
>
> Response:
>
> You are right, of course, that a novel architecture eventually needs to be proven against rich, real-world data. But, as we argue in Section 3 of the paper, we believe it would be “premature to apply the PrediNet architecture to rich, complex data before we have a basic understanding of its properties and its behaviour”. The best way to do this is with simple datasets and tasks. (We note that reviewer #1 is sympathetic to this.) The CLEVR dataset didn’t meet our requirements for reasons we have set out in Section 3 of the revised paper as follows:
>
> “Existing datasets for relational reasoning tasks, such as CLEVR (Johnson et al (2017)) and sort-of-CLEVR (Santoro et al (2017)), were ruled out because they include confounding complexities, such as occlusion and shadows or language input, and/or because they don’t lend themselves to the fine-grained task-level splits we required.”
>
> See also our response to reviewer #1.

---

### Official Review · AnonReviewer1 · 2019-10-23
**Official Blind Review #1**

**Rating:** 6

**Review:**

This paper presents PrediNet: an architecture explicitly designed to extract representations in the form of three-place predicates (relations). They evaluate the architecture on a visual relational task called the "Relations Game" which involves comparing Tetris-like shapes according to their appearance, relative positions, etc.. They show that their architecture leads to useful generalizable representations in the sense that they can be used for new tasks without retraining.

I think this paper contains a number of unusual and interesting ideas but is let down by its presentation. The writing is good, but provides very little intuition for why we should expect this approach to work (aside from its connection to equation 1) - I discuss this in more depth below. The experimental task is interesting (I'm okay with synthetic tasks of this form for unusual new architectures like this), but I'm not sure what it tests that isn't tested in the CLEVR and sort-of -CLEVR datasets which rely on similar relational reasoning to solve. The advantage of those datasets is they are well-established with strong baselines so we can be more certain that a fair comparison is been made. I've voted to reject this paper because I feel its premature in its current form.

Expanding on this - The description of PrediNet covers the basics, but missing detail and intuition for why certain choices are made. For example,
 - why is L flattened for the queries (I think it’s because the query is independent of pixel location, but flattening seems of when L also includes co-ordinates) but not the key, K?
 - Why is the key space shared between heads (this seems more intuitive - keys should have consistent meaning… but if that’s the case, make that intention explicit)?
 - Also, writing the dimensionality of the matrices, would help (e.g. is W_S in R^{m x j} or R^{m x 1} or something else?).
 - What is the meaning of the position features in E_1 and E_2? From the softmax product it seems they should be a weighted sum of the pixels that are addressed - implying that it is the weighted average location?
 - The final representation mostly consists of an h x (j) vector (ignoring positions) containing the output of the comparators. Could you provide some intuition for why we would expect such a representation to be useful for the downstream task? This representation seems to differ substantially from what is used in the baseline methods: i.e. attention-weighted sums of the input features.



**Experience Assessment:**

I have read many papers in this area.

**Review Assessment: Checking Correctness Of Derivations And Theory:**

N/A

**Review Assessment: Checking Correctness Of Experiments:**

I assessed the sensibility of the experiments.

**Review Assessment: Thoroughness In Paper Reading:**

I read the paper at least twice and used my best judgement in assessing the paper.

---

> ### Author Response · Authors · 2019-11-07
> **Response to reviewer #1 - part 1**
>
> Many thanks for the thoughtful review. We have revised the paper accordingly. We hope you feel your concerns have been addressed, and that you will consider recommending acceptance. (Note that this is a two-part response.)
>
>
> Reviewer #1: “I'm not sure what [the Relations Game] tests that isn't tested in the CLEVR and sort-of -CLEVR datasets which rely on similar relational reasoning to solve …”
>
> Response:
>
> Although it’s gratifying that the PrediNet out-performs the baselines in various respects, our primary aim in this work is not to demonstrate superior performance, but to present a novel architecture conforming to certain high-level design principles, and to explore some of its basic properties. So we wanted to use simple datasets, and were keen to avoid tasks with confounding features. This ruled out CLEVR on a couple of grounds. First, the 3D nature of the images entails confounding visual complexities such as occlusion and shadows. Second, the CLEVR task involves language input, which is another confounding factor. The sort-of-CLEVR dataset was similarly ruled out, on the second of these grounds (though not the first). A further consideration was that we needed datasets that could be split into multiple tasks in many different ways, so that we could follow the three-step protocol of pre-training , freezing, and re-training (Fig. 3 of the paper). Neither CLEVR nor sort-of-CLEVR lend themselves to this kind of task-level split. Finally, at test time we wanted to use fully held-out objects - that is to say objects with colours not seen during training and shapes not seen during training - which neither CLEVR nor sort-of-CLEVR provide for. We weren’t aware of any existing datasets meeting these criteria, so we devised a new one (the Relations Game). (Indeed, we consider the Relations Game family of tasks as one of the contributions of the paper, albeit only a minor one.)
>
> We have added the following paraphrased version of the above paragraph to Section 3 the revised paper:
>
> “Existing datasets for relational reasoning tasks, such as CLEVR (Johnson et al (2017)) and sort-of-CLEVR (Santoro et al (2017)), were ruled out because they include confounding complexities, such as occlusion and shadows or language input, and/or because they don’t lend themselves to the fine-grained task-level splits we required. Consequently, we devised a new configurable family of simple classification tasks that we collectively call the Relations Game.”
>
>
> Reviewer #1: “Why is L flattened for the queries (I think it’s because the query is independent of pixel location, but flattening seems of when L also includes co-ordinates) but not the key, K?”
>
> Response:
>
> The whole (flattened) image is used to generate queries so that what a head attends to can depend on the full (non-local) context of what’s in the image. For example, perhaps a head needs to pick out the red object in the top-left of the image because there is another red object in the bottom right of the image. By contrast, the shared key space can be determined from local information only.
>
> We have added the following paraphrased version of the above sentences to Section 2 of the revised paper:
>
> “The whole (flattened) image is used to generate queries, allowing attention masks to depend on its full (non-local) content.”
>
>
> Reviewer #1: “Why is the key space shared between heads (this seems more intuitive - keys should have consistent meaning… but if that’s the case, make that intention explicit)?”
>
> Response:
>
> Yes, this ensures that the set of entities that are candidates for attention is consistent across heads. We have added the following clause to the relevant sentence in Section 2 of the revised paper:
>
> “… so that the set of entities that are candidates for attention is consistent across heads.”

---

> ### Author Response · Authors · 2019-11-07
> **Response to Reviewer #1 - Part 2**
>
> This is the second part of our two-part response
>
> Reviewer #1: “Also, writing the dimensionality of the matrices, would help (e.g. is W_S in R^{m x j} or R^{m x 1} or something else?)”
>
> Response:
>
> The dimensionality of all the matrices is indicated in Fig. 1. As stated there, W_S is in R^{m x j}. (Sorry the font size is a bit small; there’s a lot packed in to that figure. But the resolution is high, so everything should be legible if viewed on screen.)
>
>
> Reviewer #1: “What is the meaning of the position features in E_1 and E_2? From the softmax product it seems they should be a weighted sum of the pixels that are addressed - implying that it is the weighted average location?”
>
>
> Response:
>
> The last two elements of each input feature vector in L are the xy co-ordinates of the associated patch in the image generated by the CNN. So, you are quite right that, after softmax, this yields the weighted average location. In practice, whether a head attends focally to a specific object, or diffusely to a patch of the background, this yields the centre of mass of whatever is being attended to, which is intuitive and seems to work well. (Unfortunately there isn’t space for this clarification in the revised paper.)
>
>
> Reviewer #1: “The final representation mostly consists of an h x (j) vector (ignoring positions) containing the output of the comparators. Could you provide some intuition for why we would expect such a representation to be useful for the downstream task? …”
>
> Response:
>
> Thank you for prompting us to do this. Section 7 of the revised paper has the following new material:
>
> “Although, the present experiments don’t use the fully propositional version of the PrediNet output, the concatenated vector form inherits many of its beneficial properties, notably a degree of compositionality. In particular, one important respect in which the PrediNet differs from other network architectures is the extent to which it canalises information flow; at the core of the network, information is organised into small chunks that are processed in parallel channels that limit the ways the chunks can interact. We believe this pressures the network to learn representations where each separate chunk of information (such as a single value in the vector R*) has independent meaning and utility. (We see evidence of this in the relational disentanglement of Fig. 6.) The result is a representation whose component parts are amenable to recombination, and therefore re-use in a novel task.”

---

### Official Review · AnonReviewer3 · 2019-10-24
**Official Blind Review #3**

**Rating:** 6

**Review:**

The authors propose a new neural network architecture that learns to form propositional representations with an explicitly relational structure from raw pixel data. The authors testified the proposed algorithm using the Relations Game, whose aim is to label an image containing a number of objects as True or False according to whether a given relationship holds among the objects in the image. This paper is well organized. The applied methods are introduced in detail. The authors showed the improvement using the Relations Game.

**Experience Assessment:**

I do not know much about this area.

**Review Assessment: Checking Correctness Of Derivations And Theory:**

I did not assess the derivations or theory.

**Review Assessment: Checking Correctness Of Experiments:**

I did not assess the experiments.

**Review Assessment: Thoroughness In Paper Reading:**

I made a quick assessment of this paper.

---

> ### Author Response · Authors · 2019-11-07
> **Response to Reviewer #3**
>
> Thank you for your review. (If you feel we have responded well to the other reviewers' comments, please consider raising your score.)

---

### Public Comment · ~Daniel_Azulay1 · 2020-02-07
**Question about Figure 1**

In figure 1,
the two boxes under softmax(Q1 KT)  and softmax(Q2 KT) are shown to have dimension g.
Shouldn't they have dimension n?

---

> ### Author Response · Authors · 2020-02-07
> **Question about Figure 1**
>
> Yes, thank you. This will be corrected next time we release a version of the paper

---

### Decision · Program_Chairs · 2019-12-19

**Decision:**

Reject

**Comment:**

This paper proposes a model that can learn predicates (symbolic relations) from pixels and can be trained end to end.  They show that the relations learned generate a representation that generalizes well, and provide some interpretation of the model.

Though it is reasonable to develop a model with synthetic data, the reviewers did wonder if the findings would generalize to new data from real situations.  The authors argue that a new model should be understood (using synthetic data) before it can reasonably be applied to natural data.  I hope the reviews have shown the authors which areas of the paper need further explanation, and that the use of a synthetic dataset needs to strong justification, or perhaps show some evidence that the method will probably work on real data (e.g. how it could be extended to natural images).